# A prophylactic subcutaneous dose of the anticoagulant tinzaparin does not influence qPCR-based assessment of circulating levels of miRNA in humans

**Abraham Nilsson**[1], **Anna Maria Nerhall**[2], **Ivan Vechetti**[3], **Lotta Fornander**[4], **Simon Wiklund**[2], **Björn Alkner**[2], **Jörg Schilcher**[5,6‡], **Ferdinand von Walden**[7‡]*

1 Department of Anesthesia and Intensive Care, and Department of Biomedical and Clinical Sciences, University Hospital Linköping, Linköping University, Linköping, Sweden, 2 Department of Orthopedics Eksjö, Region Jönköping County and Department of Clinical and Experimental Medicine, Linköping University, Linköping, Sweden, 3 Department of Nutrition and Health Sciences, University of Nebraska-Lincoln, Lincoln, Nebraska, United States of America, 4 Department of Orthopedic Surgery in Norrköping and Department of Biomedical and Clinical Sciences, Linköping University, Norrköping, Sweden, 5 Department of Orthopedics and Department of Biomedical and Clinical Sciences, Faculty of Health Science, Linköping University, Linköping, Sweden, 6 Wallenberg Center for Molecular Medicine, Linköping University, Linköping, Sweden, 7 Division of Pediatric Neurology, Department of Women's and Children's health, Karolinska Institute, Stockholm, Sweden

‡ JS and FW contributed equally to this work as senior authors.
* Ferdinand.von.walden@ki.se

**Data Availability Statement:** All relevant data are within the paper and its Supporting Information files.

## Abstract

Circulating microRNAs (miRNAs) have become increasingly popular biomarker candidates in various diseases. However, heparin-based anticoagulants might affect the detection of target miRNAs in blood samples during quantitative polymerase chain reaction (qPCR)-based analysis of miRNAs involving RNA extraction, cDNA synthesis and the polymerase catalyzed reaction. Because low-molecular-weight heparins (LMWH) are widely used in routine healthcare, we aimed to investigate whether a prophylactic dose of the LMWH tinzaparin influences qPCR-based quantification of circulating miRNAs. A total of 30 subjects were included: 16 fracture patients with tinzaparin treatment and 14 non-fracture controls without anticoagulation therapy. To control for the effect of tinzaparin on miRNA analysis an identical concentration of synthetic miRNAs was added to plasma, isolated RNA and prepared complementary DNA (cDNA) from all samples in both groups. No significant difference was observed for cDNA synthesis or qPCR when comparing tinzaparin-treated patients with untreated controls. Among the tinzaparin-treated patients, plasma levels of six endogenous miRNAs (hsa-let-7i-5p, hsa-miR-30e-5p, hsa-miR-222-3p, hsa-miR-1-3p, hsa-miR-133a-3p, hsa-miR-133b) were measured before and one to six hours after a subcutaneous injection of tinzaparin 4500IU. No significant effect was observed for any of the investigated miRNAs. A prophylactic dose of 4500IU tinzaparin does not seem to affect cDNA synthesis or qRT-PCR-based quantification of circulating miRNAs.

**Funding:** This work was supported by Linköping University, Region Östergötland, the Medical Research Council of Southeast Sweden (JS, grant no. FORSS-852501), Futurum – the Academy for Health and Care, Region Jönköping County, Sweden (BA, grant no. FUTURUM-937508 and 870471) and the Swedish Kidney Foundation (FvW, F2019-0048). Furthermore, we thank the Knut and Alice Wallenberg Foundation for generous support to JS. The funders had no role in study design, data collection and analysis, decision to publish, or preparation of the manuscript.

**Competing interests:** The authors have declared that no competing interests exist.

## Introduction

Biomarker-based diagnostics and monitoring of various diseases are becoming increasingly popular. Liquid biopsies (e.g., plasma and serum sampling) offer a convenient way to characterize an ongoing pathophysiological process and predict disease progression, thereby allowing for early measures to influence the course of the disease. In recent years, the analysis of circulating miRNAs in peripheral blood has been established and added to the list of important biomarker candidates [1–3].

To quantify miRNAs, quantitative (q) polymerase chain reaction (PCR) is the most established technique. However, the anticoagulant heparin and its derivates might influence qPCR-based analysis of miRNAs during RNA extraction, cDNA synthesis and the polymerase catalyzed reaction, resulting in an underestimation [4]. Initial reports identified an inhibitory effect of heparin on RNA quantification by direct interference with the DNA polymerase used in qPCR [5]. Further investigations found a dose-dependent effect of heparin on the detection of endogenous miRNAs and a C. Elegans miRNA (cel-39) spike-in during in vitro and in vivo investigations in patients undergoing heparinization as part of cardiac catheterization. Lower levels of spike-in RNAs were detected when the spike-in was added prior to RNA isolation in a heparin-containing serum sample as compared to a control sample [6]. These observations have been confirmed in a recent study in both plasma and urine [7].

Today, many patients receive thromboembolic prophylaxis including subcutaneous tinzaparin. Considering that tinzaparin has a half-life of about 3.9 hours, it can be estimated that a ~16 h time period is needed to eliminate 95% of the drug [8, 9]. Timing strategies for blood sampling in clinical routines to avoid this inhibitory effect would be difficult to implement. However, it remains unclear if a prophylactic dose of tinzaparin has similar effects on RNA isolation, cDNA synthesis and qPCR-based quantification of exogenous and endogenous miR-NAs as a therapeutic dose of heparin. Therefore, the aim of this study was to investigate the effect of a prophylactic dose of tinzaparin on the different stages of miRNA extraction in clinical blood samples.

## Methods

### Study design

This is a study using existing blood samples from patients enrolled in an ongoing study on acute compartment syndrome (BioFACTS, Clinicaltrials.gov, Identifier: NCT04674592) and from healthy control subjects enrolled in a study on the effect of exercise on mitochondria-derived peptides [10], see graphical abstract (Fig 1). These studies were approved by the Regional Ethical Review board in Linköping (Dnr. 2017/514-31 and 2017/183-31).

### Research subjects

**Tinzaparin group.** In the BioFACTS study, patients aged 15–65 years with traumatic tibial fractures are included. Exclusion criteria are malignancy, acute myocardial infarction, kidney failure (GFR ≤35 ml/min), muscle disease and paraplegia/tetraplegia. Only fracture patients with available blood samples were selected (N = 16). These samples were then divided into two subgroups depending on when the blood sample was drawn in relation to the latest dose of tinzaparin, < 16 hours (early tinzaparin group (ETG)) or > 16 hours (late-TG (LTG)). We considered a 16-hour interval as a sufficient washout time to allow elimination of tinzaparin based on its half-life of ~3.9 hours [8, 9].

**Control group.** As controls (CON), resting blood samples from healthy subjects (without any ongoing anticoagulation therapy) who participated in a study on the effects of resistance

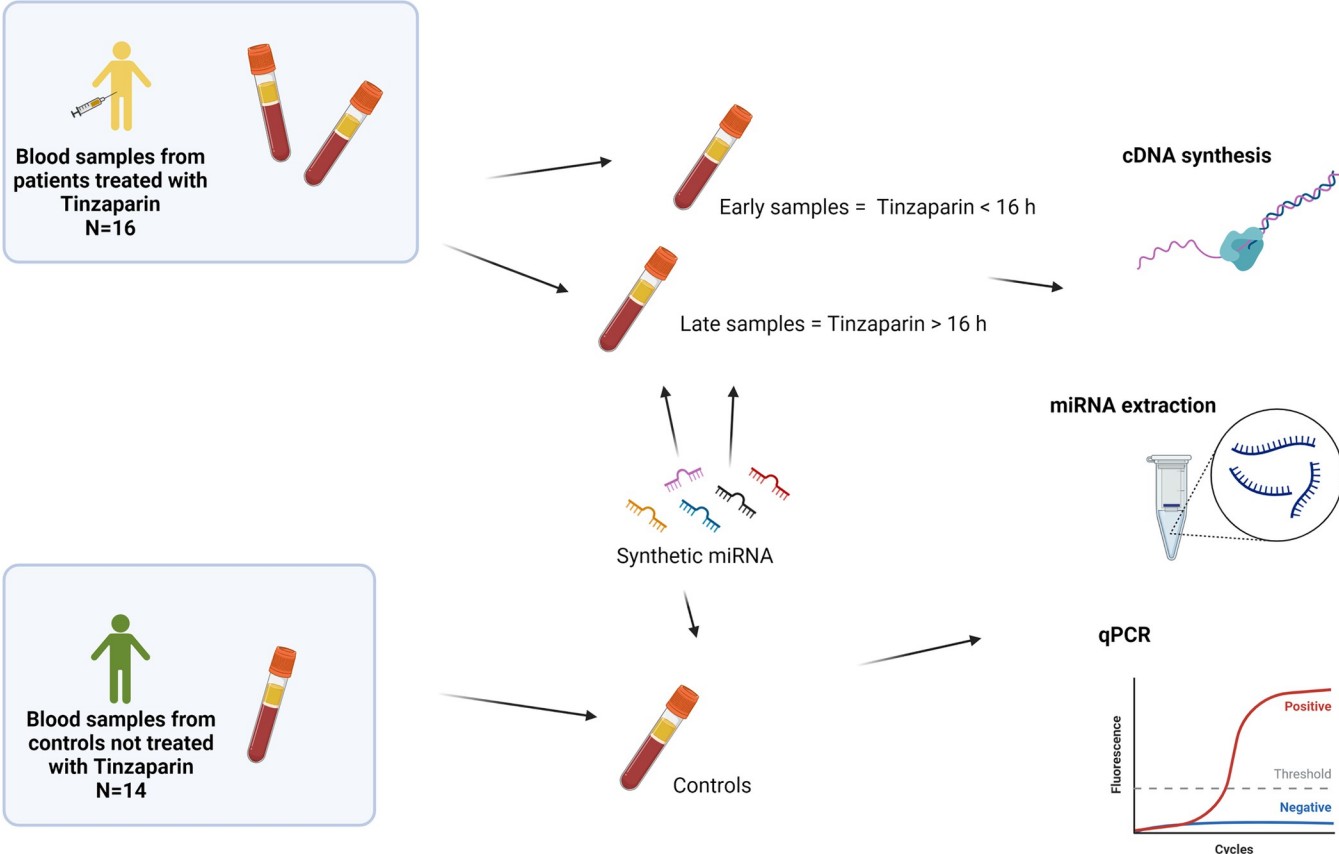

**Fig 1. Graphical abstract.** Graphical presentation of the study design. Sixteen fracture patients with tinzaparin treatment and 14 non-fracture controls without anticoagulation therapy. To control for the effect of tinzaparin on miRNA analysis an identical concentration of synthetic miRNAs was added to plasma, isolated RNA and prepared cDNA from all samples in both groups prior to qPCR.

and endurance exercise on mitochondria-derived peptides were used [10]. The research subject characteristics are shown in Table 1.

**Anticoagulant therapy.** Based on the local routine, all fracture patients were given subcutaneous tinzaparin 4500 IU once daily except patients with a body weight > 90kg, who received a two-dose regimen. None of the subjects in the control group received any anticoagulant therapy.

**Blood sampling and preparation of plasma.** In all tinzaparin-treated patients, blood samples were collected in EDTA tubes (BD, Franklin Lakes, NJ) at six-hour intervals for a maximum of 48 hours, starting on admission to the hospital. If surgery was initiated within these 48 hours, the first sample series was terminated. After surgery, blood sampling was resumed at six-hour intervals for an additional 24 hours. In the control group, blood samples

**Table 1.**

|  | Tinzaparin group N = 16 | Control group N = 14 |
|---|---|---|
| Men/women. | 10/6 | 7/7 |
| Age, years. Mean (range). | 48 (15–67) | 32 (18–50) |
| Height, meters. Mean (range). | 1.73 (155–185) | 1.73 (1.59–1.88) |
| Weight, kg. Mean (range). | 85 (52–111) | 76 (46–110) |
| BMI, kg/m$^2$. Mean (range). | 28 (22–36) | 25 (18–34) |

in EDTA tubes (BD) were drawn once at study inclusion. Within two hours after sampling, the tubes were centrifuged at 2,500 rpm for 10 minutes at room temperature. The plasma supernatant was then immediately transferred to cryo-tubes and stored at -80˚C.

**RNA extraction and cDNA synthesis.** All experiments were conducted by QIAGEN Genomic Services (Hilden, Germany). The plasma was thawed on ice and centrifuged at 3000 x g for five min in a 4˚C microcentrifuge. An aliquot of 200 μL per sample was transferred to a FluidX tube and 60 μl of Buffer RPL containing 1μg carrier-RNA per 60μl Buffer RPL and RNA spike-in template mixture (UniSp2, UniSp4 and UniSp5, QIAGEN) was added to the sample and mixed for one min and incubated for seven min at room temperature, followed by the addition of 20 μL Buffer RPP. All three spike-ins were pre-mixed at different concentrations in 100-fold increments (UniSp2>UniSP4>UniSp5). Total RNA was extracted from the samples using miRNeasy Serum/Plasma Advanced Kit; high-throughput bead-based protocol v.1 (Hilden, Germany) in an automated 96 well format. The purified total RNA was eluted in a final volume of 50 μl. Two μl RNA was reverse transcribed in 10 μl reactions using the miR-CURY LNA RT Kit (QIAGEN). To control cDNA synthesis efficiency, a fourth synthetic RNA spike-in control (UniSp6, QIAGEN) was added to the reverse transcriptase (RT) reaction mix allowing evaluation of the RT reaction. Polymerase activity was assessed by the addition of a fifth spike-in control (UniSp3, QIAGEN) added to each cDNA sample. In addition, a no template control (NTC) sample in the RT step was included as a negative control to detect potential RNA contamination in the RT step.

**qRT-PCR.** All experiments were conducted by QIAGEN Genomic Services. cDNA was diluted 50 x and assayed in 10 μl PCR reactions according to the protocol for miRCURY LNA miRNA PCR; each miRNA was assayed once by qPCR on the miRNA Ready-to-Use PCR, custom panel using miRCURY LNA SYBR Green master mix. Negative controls excluding the template from the reverse transcription reaction was performed and profiled like the samples. The amplification was performed in a LightCycler® 480 Real-Time PCR System (Roche) in 384 well plates. The amplification curves were analyzed using the Roche LC software, both for determination of Cq (by the second derivative method) and for melting curve analysis. As part of standard PCR data quantitative competitive (QC), each individual amplification product on the PCR panel was scrutinized by 1) melting curve analysis, 2) calculation of amplification efficiency and, 3) comparison of the quantification cycle (Cq) value to the background level in the negative control sample (NTC). An assay detected 5 Cq lower than the NTC was included in the data analysis. For assays that did not yield any signal on the negative control, the upper limit of detection was set to Cq = 37.

Six miRNAs were investigated based on previously observed effects of intravenous heparin treatment, on muscle specificity given the possibly of muscle injury following a tibial fracture (hsa-miR-1-3p, hsa-miR-133a-3p and hsa-miR-133b) [11] or suitability as stable circulating miRNAs (internal reference QIAGEN) hsa-let-7i-5p, hsa-miR-30e-5p, hsa-miR-222-3p.

**Statistical analysis.** Data were tested for normality using the Shapiro-Wilks test. When normally distributed, differences between groups were investigated using a one-way analysis of variance (ANOVA); otherwise non-parametric statistics were used (Kruskal-Wallis). The effect of tinzaparin on endogenous miRNA levels was investigated with a paired two-tailed t-test. Data are presented as mean±SD unless otherwise stated.

## Results

### Effect of prophylactic tinzaparin treatment on cDNA synthesis and qPCR

To assess the effect of 4500 IU tinzaparin on the cDNA synthesis reaction and polymerase activity during the qPCR reaction, we compared raw Cq values of spike-in UniSp6 and

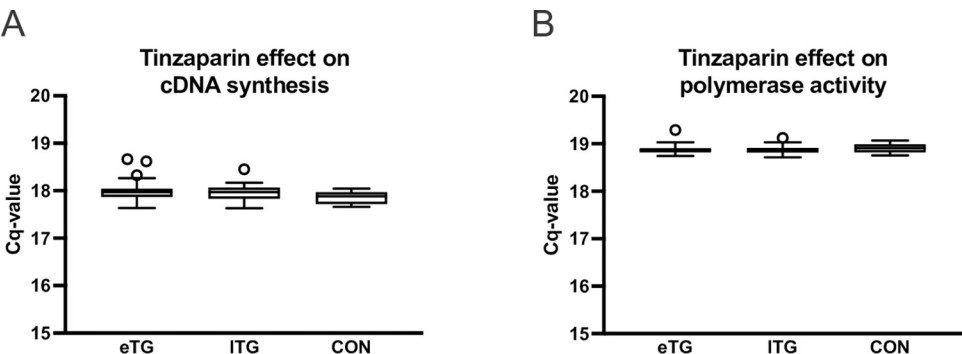

**Fig 2. Effect of prophylactic tinzaparin treatment on cDNA synthesis and qPCR.** Cq values for two synthetic spike-in RNAs, A) UniSp6 added to RNA prior to cDNA synthesis and B) UniSp3 added to cDNA prior to qPCR; UniSp6, ETG n = 71, LTG n = 58, CON n = 14; UniSp3, ETG n = 71, LTG n = 58, CON n = 14. ETG = early tinzaparin group, LTG = late tinzaparin group, CON = control. The horizontal line shows the median, while the box encompasses the 50th percentile and the whiskers encompass the minimum and maximum values. Outliers are displayed as points. Outliers are defined according to Tukey, as values exceeding more than the 75 percentile plus 1.5 IQR.

UniSp3 in plasma samples from ETG, LTG and control. No significant effect of 4500 IU tinzaparin was observed on cDNA synthesis (Cq mean±SD, UniSp6 ETG 18.0±0.2, LTG 17.9±0.2, CON 17.9±0.1, Kruskal-Wallis test p = 0.06) or qPCR chemistry (Cq mean±SD, UniSp3 ETG 18.9±0.1, LTG 18.9±0.1, CON 18.9±0.1, Kruskal-Wallis test p = 0.17, Fig 2A and 2B).

## Effect of prophylactic tinzaparin treatment on RNA extraction from plasma

To measure the effect of tinzaparin on RNA extraction, the spike-ins UniSp2, UniSp4 and UniSp5 were pre-mixed at different concentrations in 100-fold increments (UniSp2>UniSP4>UniSp5) and added to all samples prior to RNA extraction. All three spike-ins were detected at similar concentrations (Cq mean±SD; UniSp2 ETG 20.4±0.6, LTG 20.4±0.7, CON 20.0±0.6, one-way ANOVA p = 0.89; UniSp4 ETG 27.5±0.6, LTG 27.4±0.7, CON 27.1±0.5, Kruskal-Wallis test p = 0.01; UniSP5 ETG 35.2±1.0, LTG 35.3±1.0, CON 35.1±0.9, Kruskal-Wallis test p = 0.92) in all groups (Fig 3).

## Effect of prophylactic tinzaparin treatment on quantification of different endogenous circulating miRNAs

To investigate whether tinzaparin treatment acutely affects qPCR-based quantification of endogenous miRNAs we performed additional experiments on a selection of samples from the ETG. Specifically, we compared pre vs post tinzaparin administration effects on selected patients that had a pre-tinzaparin blood sample coupled with a post tinzaparin blood sample within six hours from the administered dose. This resulted in a subpopulation from ETG of n = 11. In this specific setting the fold change (FC) in miRNA abundance was investigated pre vs post tinzaparin. None of the endogenous miRNAs studied displayed a unidirectional change in FC values or a significant pre vs post FC difference (pre vs post FC, mean±SD; hsa-let-7i-5p 0.95±0.2 p = 0.41, hsa-miR-30e-5p 1.06±0.2 p = 0.29, hsa-miR-222-3p 0.95±0.1 p = 0.16, hsa-miR-1-3p 0.94±0.6 p = 0.16, hsa-miR-133a-3p 0.92±0.6 p = 0.20 and hsa-miR-133b 1.07±0.7 p = 0.57, Fig 4A–4F).

## Discussion

In the present study, we show that thrombosis prophylaxis with subcutaneous tinzaparin, 4500 IU does not seem to influence cDNA synthesis or qPCR-based quantification of endogenous

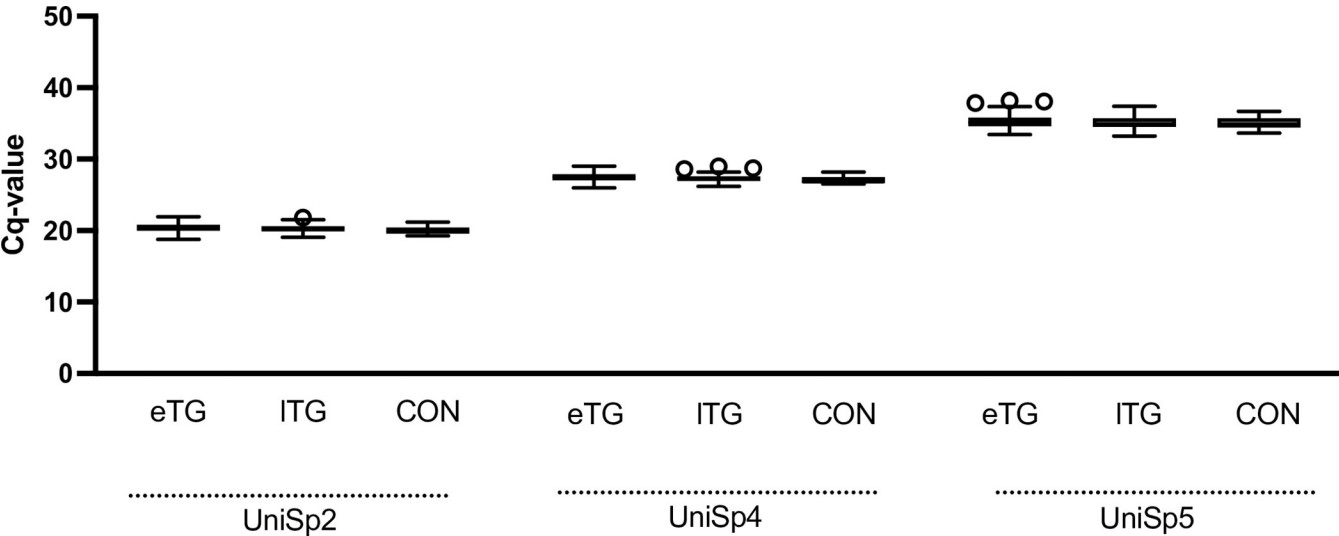

**Fig 3. Effect of prophylactic tinzaparin treatment on RNA extraction from plasma.** Cq values for three synthetic spike-in RNAs added to plasma prior to RNA extraction; UniSp2 ETG n = 71, LTG n = 57, CON n = 14; UniSp4, ETG n = 71, LTG n = 57, CON n = 14 & UniSp5, ETG N = 69, LTG n = 56, CON n = 14. ETG = early tinzaparin group, LTG = late tinzaparin group, CON = control. The horizontal line shows the median, while the box encompasses the 50th percentile and the whiskers encompass the minimum and maximum values. Outliers are displayed as points. Outliers are defined according to Tukey, as values exceeding more than the 75 percentile plus 1.5 IQR. Multiple comparisons showed a significant difference between ETG and CON, p = 0.017, denoted by *.

or spike-in miRNAs in clinical samples. These results contrast with earlier studies showing an effect of LMWH and Heparin on miRNA analysis [7, 12]. This discrepancy might be related to differences in administration (subcutaneous vs *in vitro*/intravenous) or pharmacodynamics

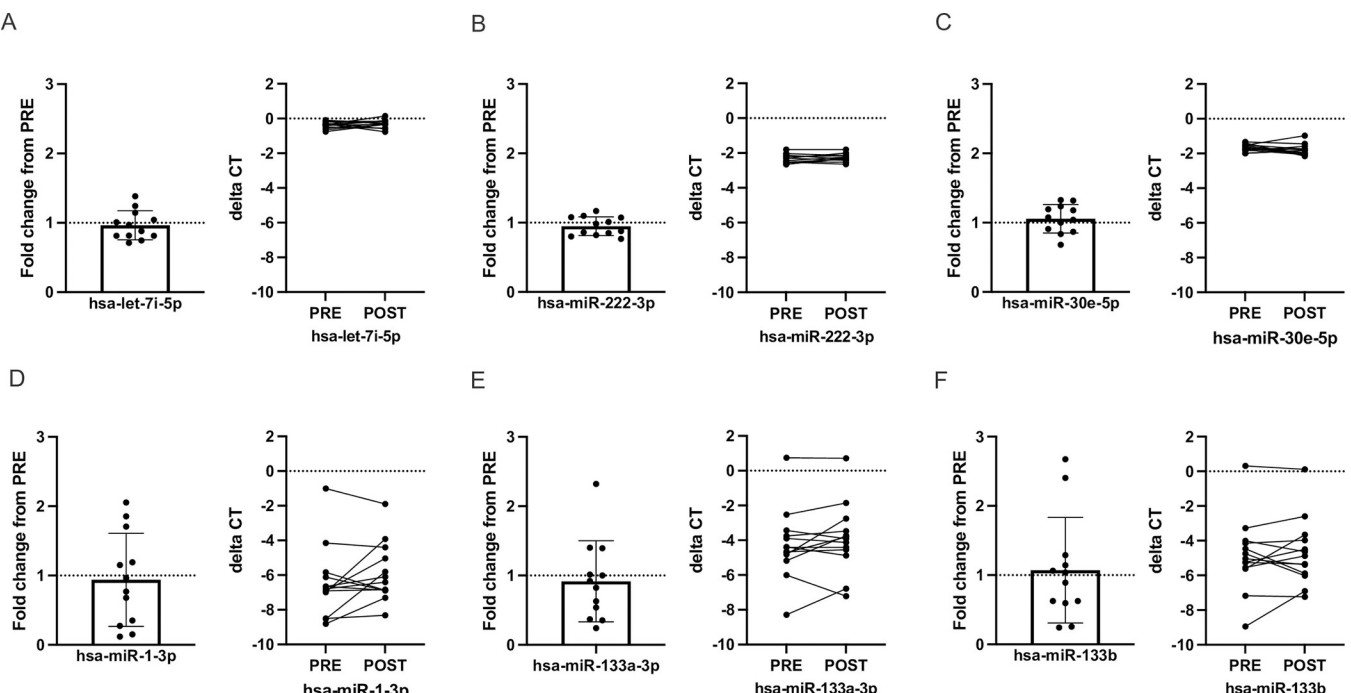

**Fig 4. Effect of prophylactic tinzaparin treatment on quantification of different endogenous circulating miRNAs in fracture patients.** Effect of a dose of tinzaparin (<6 hours) on levels of hsa-let-7i-5p (A), hsa-miR-222-3p (B), hsa-miR-30e-5p (C), hsa-miR-1-3p (D), hsa-miR-133a-3p (E) & hsa-miR-133b (F) in plasma. Data expressed as fold change from PRE (left graph) and delta CT pre vs post (right graph), mean±SD, n = 12.

and -kinetics between our and the two previously reported studies. Higher Cq values of target miRNAs in previous studies were observed during the first 35 minutes following intravenous heparin bolus of 75 IU/kg [12]. Likewise, an *in vivo* effect was confirmed in a similar study with samples collected from patients receiving an intravenous heparin bolus of 5000 IU and an additional maintenance dose of 2500–5000 IE during heart catheterization. A marked reduction in miRNA levels was detected 10 minutes after heparin administration, and persisted at the 60-minute time point [6]. The half-life after an intravenous injection of heparin 100 IU/kg is ~1 hour [13] and results in a high and rapidly transient heparin concentration in plasma, which might explain the inhibitory effect on miRNA analysis seen in earlier studies. Unlike heparin, tinzaparin is administered subcutaneously. This leads to a lower peak concentration in plasma and a longer half-life for tinzaparin, about 3.9 hours [8, 9].

To investigate whether tinzaparin treatment influences spin column-based RNA extraction, three different spike-in RNAs were added at incremental concentrations. We observed a minor yet statistically significant effect for the mid-level concentration spike-in RNA (UniSp4) whereas the high (UniSp2) and low concentration (UniSp5) spike-ins were unaffected by tinzaparin administration. Spin column-based RNA isolation is based on electrostatic interactions between the silica membrane of the column and the negatively charged RNA. The negative charge of heparin has been suggested as a potential reason for co-extraction with RNA [14]. Our experimental setup does not allow us to elucidate the underlying mechanism behind this effect on RNA isolation but we conclude that it is minor and that it probably does not interfere with downstream analysis.

When we investigated the effect of tinzaparin on RNA extraction, cDNA synthesis and qPCR chemistry, we considered the effect of tinzaparin as negligible if the sample was collected more than 16 hours after administration. This time interval was based on calculations of the well-established paradigm that four half-life cycles typically lead to the elimination of roughly 95% of a drug from serum. In our study, blood samples were drawn irrespective of the administration of tinzaparin. While this might lead to the reduction of systematic errors, it also limited the number of samples (34 out of 129) that were collected close to the peak of anti-Xa activity, (two to six hours after administration). Thus, most samples were collected on the up-/downslope, which might have influenced our results.

To further characterize the potential negative effect of a prophylactic dose of tinzaparin on qPCR-based quantification of miRNAs we assessed the levels of six endogenous miRNAs in plasma including three muscle-specific miRNAs [3, 15]. Previous studies have reported effects on measurement levels of circulating miRNAs after an intravenous heparin bolus of 75 IU/kg [12]. These effects were also seen *ex vivo* by adding heparin or LMWH to either plasma or urine samples prior to miRNA quantification [6, 7]. None of the investigated endogenous miRNAs was affected by the tinzaparin treatment. This supports our interpretation that 4500 IU of tinzaparin does not influence qPCR-based quantification of plasma miRNAs. Although not significantly influenced by the injection of tinzaparin, all muscle-specific miRNAs (Fig 4D and 4E) showed a larger variability pre vs post tinzaparin administration compared to non-muscle-specific miRNAs (Fig 4A–4C). This variability might be related to the tibial fracture and/or the surgical treatment of it rather than the tinzaparin treatment. Traumatic fractures are always associated with soft tissue damage, and skeletal muscle-specific miRNAs are increased in peripheral blood as a result [15]. Our samples were collected at different time points after the fracture, which might have contributed to the bidirectional spread in fold change values for the skeletal muscle-specific miRNAs.

Patient characteristics differ slightly between the groups, most notably in age. The pharmacokinetics of tinzaparin however, only seem to be affected by severe renal impairment in adults and not by age or gender [16]. In the BioFACTS study, renal impairment (GFR < 35 ml/min)

is an exclusion criterion and all patients in our control group were healthy. Therefore, we believe it is unlikely that differences in pharmacokinetics of tinzaparin would affect our analyses. Endogenous miRNA levels, were compared between two time points within the same patient, which precludes any effect of age or gender differences between the groups. Our results are important for clinical scientists conducting research on miRNAs in clinical settings. Currently, many hospitalized patients receive venous thromboembolism prophylaxis, often with a similar regimen as in the current study. Moreover, the ongoing Sars-Cov2 pandemic has contributed to an increase in the number of LMWH-treated patients, given the strong treatment recommendations for thromboembolism prophylaxis [17]. Our results should be reassuring for clinical scientists, indicating that miRNA levels seem unaffected by a prophylactic subcutaneous dose of tinzaparin.

## Limitations

Our study is limited by its small sample size and differences in patient characteristics between groups. Most notably, all patients with tinzaparin treatment also had a tibial fracture, while none of the controls had fractures performed in quite small, 16 patients in the study group and 14 in the control group, which adds some uncertainty to the results. These limitations should be mitigated by our study design comparing not only patients with healthy controls but also using patients as their own controls by comparing blood samples from different time points.

## Conclusion

In the present study we show that thrombosis prophylaxis with tinzaparin 4500 IE subcutaneously does not seem to interfere with qPCR-based detection and quantification of circulating miRNAs in plasma. These results should be reassuring, as prophylactic LMWH administration is considered standard care in many hospital settings to prevent thromboembolic events.

## Supporting information

**S1 Dataset.**
(XLSX)

## Acknowledgments

We wish to thank Jörg Krummheuer, Dr. rer. nat. and QIAGEN Genomic Services for their assistance with sample preparation and the miRNA analysis. The graphical abstract was generated using BioRender.

## Author Contributions

**Conceptualization:** Abraham Nilsson, Ivan Vechetti, Björn Alkner, Jörg Schilcher, Ferdinand von Walden.

**Data curation:** Abraham Nilsson, Anna Maria Nerhall, Lotta Fornander, Simon Wiklund, Ferdinand von Walden.

**Formal analysis:** Anna Maria Nerhall, Simon Wiklund, Ferdinand von Walden.

**Funding acquisition:** Lotta Fornander, Björn Alkner, Jörg Schilcher, Ferdinand von Walden.

**Investigation:** Björn Alkner, Jörg Schilcher, Ferdinand von Walden.

**Methodology:** Ivan Vechetti, Jörg Schilcher, Ferdinand von Walden.

**Project administration:** Abraham Nilsson, Lotta Fornander, Björn Alkner, Jörg Schilcher.

**Resources:** Jörg Schilcher.

**Supervision:** Jörg Schilcher, Ferdinand von Walden.

**Writing – original draft:** Abraham Nilsson, Ferdinand von Walden.

**Writing – review & editing:** Abraham Nilsson, Anna Maria Nerhall, Ivan Vechetti, Lotta Fornander, Simon Wiklund, Björn Alkner, Jörg Schilcher, Ferdinand von Walden.

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
