## [Editor Report · Decision Letter 0]

14 Feb 2022

PONE-D-21-38473A prophylactic dose of Tinzaparin does not influence qPCR-based assessment of circulating levels of miRNA in humansPLOS ONE

Dear Dr. von Walden,

Thank you for submitting your manuscript to PLOS ONE. After careful consideration, we feel that it has merit but does not fully meet PLOS ONE’s publication criteria as it currently stands. Therefore, we invite you to submit a revised version of the manuscript that addresses the points raised during the review process.

Specifically:  After reviewing the manuscript, I have several concerns regarding the presentation of the manuscript that I deemed not suitable for review at its present form (see specific comments below). One of the main concerns is the authors does not clearly explain the results in a comprehensive way which gave the impression that the results presented were very preliminary and does not meet the publication criteria.

Specific comments:

Introduction reads like a run-on paragraph, suggest breaking into three paragraphs with line 67 – 72 as first paragraph, 72 – 82 as second paragraph, 82 – 90 as third paragraph. Paragraph breaks will provide readers better thought process regarding the rationale of the study.Methods:  Line 165 – 166 belongs to the results section. Suggest moving line 165 – 166 as the opening sentences for *Effect of prophylactic Tinzaparin treatment on RNA extraction from plasma.  *Line 166 – 171 should be at the beginning of *Effect of prophylactic Tinzaparin treatment on cDNA synthesis and qPCR.  *While line 171 – 174 should lead the results of *Effect of prophylactic Tinzaparin treatment on quantification of different endogenous circulating miRNAs.*Results:  This section was very sparse, the authors need to interpret the results in a more comprehensible way and guide the readers through and expand explanation of every table, every figure, and every scheme. For example, explain the concentration difference of three uniSp2, 4, 5 used and the outlier on Fig. 2 Please submit your revised manuscript by Mar 31 2022 11:59PM. If you will need more time than this to complete your revisions, please reply to this message or contact the journal office at plosone@plos.org. Please include the following items when submitting your revised manuscript:A rebuttal letter that responds to each point raised by the academic editor and reviewer(s). You should upload this letter as a separate file labeled 'Response to Reviewers'.A marked-up copy of your manuscript that highlights changes made to the original version. You should upload this as a separate file labeled 'Revised Manuscript with Track Changes'.An unmarked version of your revised paper without tracked changes. You should upload this as a separate file labeled 'Manuscript'.

We look forward to receiving your revised manuscript.

Kind regards,

Baochuan Lin, Ph.D.

Academic Editor

PLOS ONE

Journal Requirements:

"This work was supported by Linköping University, Region Östergötland, the Medical Research Council of Southeast Sweden (JS, grant no. FORSS-852501), Futurum – the Academy for Health and Care, Region Jönköping County, Sweden (BA, grant no. FUTURUM-937508 and 870471) and the Swedish Kidney Foundation (FvW, F2019-0048). Furthermore, we thank the Knut and Alice Wallenberg Foundation for generous support to JS. The funders had no role in study design, data collection and analysis, decision to publish, or preparation of the manuscript."

We note that you have provided funding information. However, funding information should not appear in the Funding section or other areas of your manuscript. We will only publish funding information present in the Funding Statement section of the online submission form. 

"This work was supported by Linköping University, Region Östergötland, the Medical Research Council of Southeast Sweden (JS, grant no. FORSS-852501), Futurum – the Academy for Health and Care, Region Jönköping County, Sweden (BA, grant no. FUTURUM-937508 and 870471) and the Swedish Kidney Foundation (FvW, F2019-0048). Furthermore, we thank the Knut and Alice Wallenberg Foundation for generous support to JS. The funders had no role in study design, data collection and analysis, decision to publish, or preparation of the manuscript."
---

## [Author Response · Author response to Decision Letter 0]

30 Mar 2022

Dear Dr Lin,

We hereby resubmit our manuscript and hope that the amended version is deemed satisfactory and found suitable to be sent out for review.

We have reworked the manuscript accordingly to your suggestions. Specifically, we have broken up the introduction section into three parts and made the result section more detailed. We have also removed the funding statement from the manuscript file. Please find the minimal data set file uploaded in the online submission system.

All the best

Ferdinand Von Walden

---

## [Decision Letter · Decision Letter 1]

28 Jun 2022

PONE-D-21-38473R1A prophylactic dose of Tinzaparin does not influence qPCR-based assessment of circulating levels of miRNA in humansPLOS ONE

Dear Dr. von Walden,

Thank you for submitting your manuscript to PLOS ONE. After careful consideration, we feel that it has merit but does not fully meet PLOS ONE’s publication criteria as it currently stands. Therefore, we invite you to submit a revised version of the manuscript that addresses the points raised during the review process. Both reviewers agreed that your paper addresses an interesting question, however, they both stated several concerns about your study and did not recommend publication in its present form.  Please see reviewers' insightful comments below.

We look forward to receiving your revised manuscript.

Kind regards,

Baochuan Lin, Ph.D.

Academic Editor

PLOS ONE

Reviewers' comments:

Reviewer's Responses to Questions

**Comments to the Author**

1. If the authors have adequately addressed your comments raised in a previous round of review and you feel that this manuscript is now acceptable for publication, you may indicate that here to bypass the “Comments to the Author” section, enter your conflict of interest statement in the “Confidential to Editor” section, and submit your "Accept" recommendation.

Reviewer #1: (No Response)

Reviewer #2: (No Response)

2. Is the manuscript technically sound, and do the data support the conclusions?

Reviewer #1: Partly

Reviewer #2: Partly

3. Has the statistical analysis been performed appropriately and rigorously? 

Reviewer #1: No

Reviewer #2: N/A

4. Have the authors made all data underlying the findings in their manuscript fully available?

Reviewer #1: Yes

Reviewer #2: Yes

5. Is the manuscript presented in an intelligible fashion and written in standard English?

Reviewer #1: No

Reviewer #2: Yes

6. Review Comments to the Author

Reviewer #1: The authors of the manuscript entitled: “A prophylactic dose of Tinzaparin does not influence qPCR-based assessment of circulating levels of miRNA in humans” report the results of the study that aimed to investigate whether a prophylactic dose of the LMWH Tinzaparin influences qPCRbased quantification of circulating miRNAs in blood samples of patients enrolled previously into an ongoing study on acute compartment syndrome (BioFACTS, Clinicaltrials.gov, Identifier: NCT04674592).

In the group of 16 patients with traumatic tibial fractures on tinzaparin (vs. 16 of healthy controls no significant difference was observed for cDNA synthesis or qPCR when comparing tinzaparin treated patients with untreated controls. Among the tinzaparin treated patients, plasma levels of six endogenous miRNAs (hsa-let-7i-5p, hsa-miR-30e-5p, hsa-miR-222-3p, hsa-miR-1-3p, hsa-miR-133a-3p, hsa-miR-133b) were measured before and 1–6 hours after a subcutaneous injection of Tinzaparin 4500IU revealing no significant effect for any of the investigated miRNAs.

The topic is of interest, however, there are some relevant issues related to the manuscript.

In this small sample size study (groups 16 vs 16 patients) the large difference in the age of patients in the study and control group (tinazaparin 48 (15-67) years, control 32 (18-50) appears problematic, the authors should have at least attempted to adjust their analyzes for some baseline confounders.

This is further exacerbated by the fact that the patients in the study group were sampled at different timepoints after the fracture.

How can the differences be attributed directly to the fact of tinzaparin therapy ? What could be the impact of comorbidities?

The authors should further explain the rationale for selecting the described group of microRNA for analysis in their study.

The methodology of sample and microRNA analysis appears appropriate and sufficiently described.

The authors state at the end of the discussion “Our results are important for clinical scientist conducting biomarker-focused research in clinical settings. As of today, many hospitalized patients receive venous thromboembolism prophylaxis, often with a similar regimen as in the current study. Moreover, the ongoing Sars-Cov2 pandemic has contributed to an increase in LMWH-treated patients, given the strong treatment recommendations for thromboembolism prophylaxis [19]. Our results provide guidance when miRNA analysis is performed in patients on Tinzaparin treatment.”

The authors should more clearly and specifically explain to the reader why their results are reportedly “important for clinical scientists”.

The authors should explicitly list the limitations of their study in the discussion section.

A serious linguistic editing is required before the manuscript could be considered for further review and consideration for publication.

Reviewer #2: The topic of the study is interesting. It focuses on an important aspect that should be considered when miRNAs are studied from plasma.

However, I cannot understand some points.

First, UniSp2-4-5 have been added before RNA extraction to verify that Tinzaparin does not affect this step (as demonstrated in Fig.2). Then, UniSp6 and UniSp5 have been used to test cDNA synthesis step. Don’t UniSp2-4-5 already answer to this question? Indeed, since with the first step the authors conclude that Tinzaparin treatment does not affect RNA extraction also the following steps are included. Vice versa, cDNA synthesis step would be analyzed separately in the case of Tinzaparin treatment would affect the first result. To rescue the data about the cDNA synthesis, maybe, the data could be presented in a different order: first cDNA synthesis then RNA extraction.

Finally, in fig. 4 miR-1-3p, miR-133a-3p and miR133b are characterized by a high standard deviation. A different graph for all the six miRNAs should be also presented. Particularly, the segment that joins the first point to the second point should be shown for each patient.

7. PLOS authors have the option to publish the peer review history of their article (what does this mean?). If published, this will include your full peer review and any attached files.

Reviewer #1: No

Reviewer #2: **Yes: **Vitiello Marianna

---

## [Author Response · Author response to Decision Letter 1]

9 Sep 2022

Response to the reviewers

Reviewer #1: The authors of the manuscript entitled: “A prophylactic dose of Tinzaparin does not influence qPCR-based assessment of circulating levels of miRNA in humans” report the results of the study that aimed to investigate whether a prophylactic dose of the LMWH Tinzaparin influences qPCRbased quantification of circulating miRNAs in blood samples of patients enrolled previously into an ongoing study on acute compartment syndrome (BioFACTS, Clinicaltrials.gov, Identifier: NCT04674592).

In the group of 16 patients with traumatic tibial fractures on tinzaparin (vs. 16 of healthy controls no significant difference was observed for cDNA synthesis or qPCR when comparing tinzaparin treated patients with untreated controls. Among the tinzaparin treated patients, plasma levels of six endogenous miRNAs (hsa-let-7i-5p, hsa-miR-30e-5p, hsa-miR-222-3p, hsa-miR-1-3p, hsa-miR-133a-3p, hsa-miR-133b) were measured before and 1–6 hours after a subcutaneous injection of Tinzaparin 4500IU revealing no significant effect for any of the investigated miRNAs.

The topic is of interest, however, there are some relevant issues related to the manuscript.

In this small sample size study (groups 16 vs 16 patients) the large difference in the age of patients in the study and control group (Tinzaparin 48 (15-67) years, control 32 (18-50) appears problematic, the authors should have at least attempted to adjust their analyzes for some baseline confounders.

This is further exacerbated by the fact that the patients in the study group were sampled at different timepoints after the fracture.

How can the differences be attributed directly to the fact of tinzaparin therapy ? What could be the impact of comorbidities?

Response: We thank the reviewer for this insightful comment. In our understanding of the current literature, the pharmacokinetics of tinzaparin are not affected by age or gender while severe renal impairment (GFR < 30 ml/min), may reduce clearance (1). In the BioFACTS study, renal impairment (GFR < 35 ml/min) along with other diseases affecting muscle tissue, are exclusion criteria and all patients in our control group were healthy. Therefore, we believe it is unlikely that differences in pharmacokinetics of tinzaparin would affect our analyses. Also, we compare miRNA levels between two time-points but within the same patient. This design should preclude any effect of age or gender differences between the groups. This is clarified under “Discussion”, page 15 row 265-272. Unfortunately, according to our statistical advisor, our sample size precludes comparisons stratified for several covariates. These limitations, along with the inherent limitations related to our small sample size, are now clarified under “Limitations”, page 15 row 281. 

The authors should further explain the rationale for selecting the described group of microRNA for analysis in their study.

Response: We have added further clarifications to the selection of miRNAs in the amended version of the manuscript, page 10 row 170. This part of the methods section now reads:

“Six miRNAs were investigated based on previously observed effects of intravenous heparin treatment (2), of muscle specificity given the possibly of muscle injury following a tibial fracture (hsa-miR-1-3p, hsa-miR-133a-3p and hsa-miR-133b) (3) or suitability as stable circulating miRNAs (internal reference Qiagen); hsa-let-7i-5p, hsa-miR-30e-5p, hsa-miR-222-3p.”

The methodology of sample and microRNA analysis appears appropriate and sufficiently described.

The authors state at the end of the discussion “Our results are important for clinical scientist conducting biomarker-focused research in clinical settings. As of today, many hospitalized patients receive venous thromboembolism prophylaxis, often with a similar regimen as in the current study. Moreover, the ongoing Sars-Cov2 pandemic has contributed to an increase in LMWH-treated patients, given the strong treatment recommendations for thromboembolism prophylaxis [19]. Our results provide guidance when miRNA analysis is performed in patients on Tinzaparin treatment.”

The authors should more clearly and specifically explain to the reader why their results are reportedly “important for clinical scientists”.

Response: The last paragraph of the discussion is now rewritten to explain the importance of the result. “Our results should be reassuring for clinical scientist, indicating that miRNA levels seem unaffected by a prophylactic subcutaneous dose of Tinzaparin. Page 15, row 278-280.

The authors should explicitly list the limitations of their study in the discussion section.

Response: A “Limitations” heading in now added in the end of the discussion. Page 15, row, 281-291.

A serious linguistic editing is required before the manuscript could be considered for further review and consideration for publication.

Response: The amended version of the manuscript has been revised for language by ”Anchor English” (https://www.anchorenglish.com/).

Reviewer #2: The topic of the study is interesting. It focuses on an important aspect that should be considered when miRNAs are studied from plasma.

However, I cannot understand some points.

First, UniSp2-4-5 have been added before RNA extraction to verify that Tinzaparin does not affect this step (as demonstrated in Fig.2). Then, UniSp6 and UniSp5 have been used to test cDNA synthesis step. Don’t UniSp2-4-5 already answer to this question? Indeed, since with the first step the authors conclude that Tinzaparin treatment does not affect RNA extraction also the following steps are included. Vice versa, cDNA synthesis step would be analyzed separately in the case of Tinzaparin treatment would affect the first result. To rescue the data about the cDNA synthesis, maybe, the data could be presented in a different order: first cDNA synthesis then RNA extraction.

Response: We thank the reviewer for the opportunity to make this part of the result section clearer to the reader. The effects of tinzaparin could potentially influence both RNA extraction efficiency and cDNA synthesis and by only using UniSp2-4-5 spike-ins we would not be able to distinguish between these two steps. However, based on the recommendations provided by the reviewer, we have switched the order of the result segments so that the investigation on the effects on cDNA synthesis precedes the segment including the effect on RNA extraction. We hope this will make our methodology clearer.

Finally, in fig. 4 miR-1-3p, miR-133a-3p and miR133b are characterized by a high standard deviation. A different graph for all the six miRNAs should be also presented. Particularly, the segment that joins the first point to the second point should be shown for each patient.

Response: We thank the reviewer for bringing up a valuable point. We have remade the graphs so that they in addition to mean delta fold change values include individual data points and lines to join the pre and post values.

1. Hoy SM, Scott LJ, Plosker GL. Tinzaparin sodium: a review of its use in the prevention and treatment of deep vein thrombosis and pulmonary embolism, and in the prevention of clotting in the extracorporeal circuit during haemodialysis. Drugs. 2010;70(10):1319-47.

2. Jes-Niels B. Heparin Selectively Affects the Quantification of MicroRNAs in Human Blood Samples. Clin Chem. 2013;59(7):1125-6.

3. Horak M, Novak J, Bienertova-Vasku J. Muscle-specific microRNAs in skeletal muscle development. Dev Biol. 2016;410(1):1-13.

---

## [Decision Letter · Decision Letter 2]

19 Oct 2022

A prophylactic subcutaneous dose of the anticoagulant tinzaparin does not influence qPCR-based assessment of circulating levels of miRNA in humans

PONE-D-21-38473R2

Dear Dr. von Walden,

We’re pleased to inform you that your manuscript has been judged scientifically suitable for publication and will be formally accepted for publication once it meets all outstanding technical requirements.

Kind regards,

Baochuan Lin, Ph.D.

Academic Editor

PLOS ONE

Additional Editor Comments (optional):

Reviewers' comments:

Reviewer's Responses to Questions

**Comments to the Author**

1. If the authors have adequately addressed your comments raised in a previous round of review and you feel that this manuscript is now acceptable for publication, you may indicate that here to bypass the “Comments to the Author” section, enter your conflict of interest statement in the “Confidential to Editor” section, and submit your "Accept" recommendation.

Reviewer #1: All comments have been addressed

Reviewer #2: All comments have been addressed

2. Is the manuscript technically sound, and do the data support the conclusions?

Reviewer #1: Yes

Reviewer #2: Yes

3. Has the statistical analysis been performed appropriately and rigorously? 

Reviewer #1: Yes

Reviewer #2: Yes

4. Have the authors made all data underlying the findings in their manuscript fully available?

Reviewer #1: Yes

Reviewer #2: Yes

5. Is the manuscript presented in an intelligible fashion and written in standard English?

Reviewer #1: Yes

Reviewer #2: Yes

6. Review Comments to the Author

Reviewer #1: The authors sufficiently addressed all the issues raised in the review. I suggest to accept the manuscript.

Reviewer #2: the authors have adequately addressed my comments raised in a previous round of review. the manuscript is now acceptable for publication.

7. PLOS authors have the option to publish the peer review history of their article (what does this mean?). If published, this will include your full peer review and any attached files.

Reviewer #1: **Yes: **Dominika Klimczak-Tomaniak

Reviewer #2: **Yes: **Marianna Vitiello

---

## [Editor Report · Acceptance letter]

25 Oct 2022

PONE-D-21-38473R2 

A prophylactic subcutaneous dose of the anticoagulant tinzaparin does not influence qPCR-based assessment of circulating levels of miRNA in humans 

Dear Dr. von Walden:

I'm pleased to inform you that your manuscript has been deemed suitable for publication in PLOS ONE. Congratulations! Your manuscript is now with our production department. 

Kind regards, 

on behalf of

Dr. Baochuan Lin 

Academic Editor

PLOS ONE